# Sensor Fault Reconstruction based on Adaptive Sliding Mode Observer for Forklift Fault-Tolerant Control System

**Zhilu Zhang** and **Benxian Xiao** *

School of Electrical Engineering and Automation, Hefei University of Technology, Hefei 230009, China;
zhangzhilu95@163.com
* Correspondence: xiaobenxian@hfut.edu.cn

**Abstract:** For the problem of multiple sensor fault detection and reconstruction in the forklift fault-tolerant control system, a sliding mode observer (SMO) with adaptive regulation law is proposed. Based on the three-degree-of-freedom (3-DOF) model of forklift, a linear state equation with output disturbance is designed as its equivalent sensor fault model. The sensor fault is converted into an actuator fault by defining an auxiliary state variable as an output signal filter. Then the SMO-based method of sensor fault detection and reconstruction is given. Without knowing the upper bound of an unknown fault, an adaptive sliding mode observer (ASMO) can also be effective through the adaptive algorithm. Finally, experimental results further verify the effectiveness of the method, and provide a foundation for forklift fault-tolerant control.

**Keywords:** sensor fault reconstruction; output disturbance; sliding mode observer; adaptive algorithm

## 1. Introduction

With the development of modern forklift technology, handling stability and active safety are becoming increasingly important. Forklift sensors of yaw rate, sideslip angle, and roll rate are used to measure dynamic characteristics of a forklift. Deviation of these sensors' signals may lead to spurious triggering or misoperation of the control system. Once a fault occurs, it causes serious consequences [1–3], so the possibility to diagnose and reconstruct potential faults of these sensors is the key issue of safety and stability of a forklift. To detect an incipient fault as early as possible to avoid serious damage to the controlled system, a fuzzy relational sliding mode observer (FRSMO) is proposed [4].

In recent years, different types of observers have been used for sensor fault detection and reconstruction. A sliding mode observer (SMO) is commonly used as a robust control solution for mode errors and measurement disturbances. For a class of uncertain time-delay systems, a robust SMO, which guarantees estimation error stability through the Lyapunov—Krasovskii function and the Linear Matrix Inequality (LMI) optimization, has been proposed [5,6]. An SMO with two discontinuous terms was designed to compensate for fault behavior and disturbance variation from the system state estimation. It ensured robust fault reconstruction and estimation for a class of nonlinear systems described by the Takagi–Sugeno model with unmeasurable premise variables subject to the faults affecting actuators, sensor faults, and unknown disturbances [7].

In this study, an SMO with adaptive regulation law is proposed. It establishes a linear state equation with output disturbance to convert sensor faults into actuator faults. Zhao Kaihui et al. also transformed sensor faults into actuator faults by introducing a new state variable [8]. By considering the sensor fault and output disturbance as auxiliary state vectors, a robust sliding mode descriptor

observer is discussed for estimating states and reconstructing actuator and sensor faults simultaneously in the presence of uncertainties, measurement noise, and output disturbances [9,10].

For practical application in engineering, a neural adaptive observer has been developed to detect faults in sensors and actuators of unmanned aerial vehicle (UAV) systems. In the design, weighting parameters of the Neural Network (NN) were updated by using the Extended Kalman Filter (EKF). The proposed fault detection strategy was applied to a nonlinear dynamic model of UAV for evaluation [11]. Mohand Arab Djeziri et al. also dealt with the embedded fault detection and isolation (FDI) / fault-tolerant control approach applied to an over-actuated electric vehicle [12]. A new scheme based on a nonlinear dynamic model is also designed for sensor fault detection and isolation (FDI) in a UAV system. In the proposed design, the NN is used as an observer for faults of UAV sensors, and weighting parameters of the NN are updated by using the EKF [13]. Fault-tolerant control (FTC) for vehicle Steer-by-Wire (SBW) systems in the presence of an actuator fault and bounded uncertainties has been proposed. A fault observer was developed to assess fault information and faulty SBW systems. The minimax model predictive control in the delta domain was deployed to achieve tracking performance in the event of an actuator fault, system uncertainties, and disturbances [14,15]. Yosra Rkhissi-Kammoun et al. focused on single and multiple current FDI sensors for induction motor drives in an electric vehicle; two models based on diagnostic methods were proposed; the first one was a differential algebraic estimation of fault dynamics and the second one was based on a combination of the Robust Integral Sign of the Error observer [16]. Lie et al. investigated the adaptive FTC and fault reconstruction problem for autonomous underwater vehicles (AUVs) subject to ocean current disturbance and modeling uncertainty [17]. For fault diagnosis of wind turbine systems, Silvio Simani and Paolo Castaldi presented viable fault detection and isolation techniques which rely on fault estimation using data-driven methods and effective tools for managing partial knowledge of system dynamics, as well as noise and disturbance effects [18].

In contrast to previous publications, the observer designed in this study is not related to the object model, and the method is universal. With respect to fault reconstruction in the framework of FTC, an improved second-order SMO is designed to estimate the thruster fault effect. Considering the characteristics of the observer and the fact that weight of the forklift cargo changes with time, an adaptive sliding mode observer (ASMO) based on the forklift sensor fault model with output disturbance is proposed. It solves the problem that some fault reconstruction methods demand the upper bound value of the fault. According to the adaptive algorithm, adjustable gain parameters are adjusted adaptively without knowing the upper bound of the unknown fault, and the output disturbance can also be limited. The method is fairly robust and is more in accordance with the actual situation.

The paper has the following structure: Section 2 presents the three-degree-of-freedom (3-DOF) forklift model and the sensor fault model; Section 3 proposes the ASMO; in Sections 4 and 5, sensor fault reconstruction and its improvement are presented; in Section 6, validation of the experiment is given; finally, the discussion and the conclusions are given.

## 2. Forklift Model

### 2.1. 3-DOF Forklift Model

Considering the complexity of modeling, the forklift dynamics model is simplified to a 3-DOF vehicle model, i.e., the roll motion around the x-axle, the lateral motion along the y-axle, and the yaw motion around the z-axle, with the horizon as the x-o-y plane, and the vertical direction as the *z*-axis. The 3-DOF forklift model is shown in Figure 1.

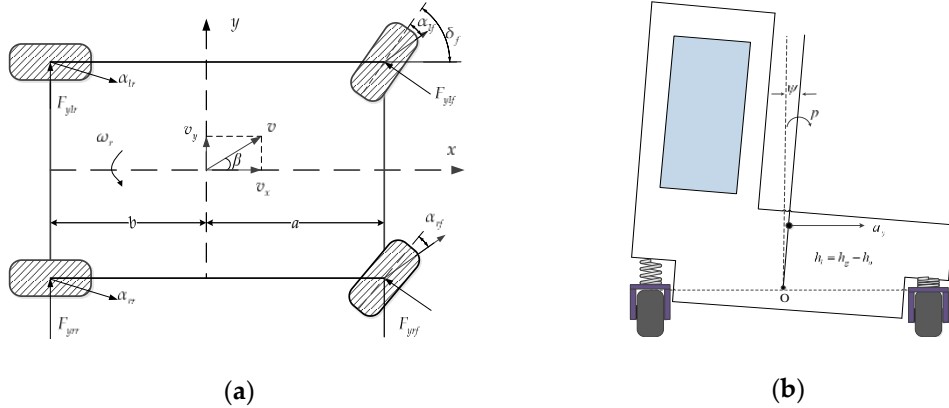

**Figure 1.** Three-degree-of-freedom (3-DOF) forklift model: (**a**) Top view; (**b**) Rear view.

The dynamic forklift model is given as follows [19]:

$$
\begin{cases}
I_x \dot{p} - I_{xz} \dot{\omega}_r = M_s g h_l \psi - K_\psi \psi - C_\psi p + M_s h_l v_x (\dot{\beta} + \omega_r) \\
M v_x (\dot{\beta} + \omega_r) - M_s h_l \dot{p} = F_{ylf} + F_{yrf} + F_{ylr} + F_{yrr} \\
I_z \dot{\omega}_r - I_{xz} \dot{p} = -b(F_{ylf} + F_{yrf}) + a(F_{ylr} + F_{yrr}) \\
F_y = k\alpha
\end{cases}
\tag{1}
$$

where $M$ is the total mass of the vehicle, $M_s$ is the sprung mass of the vehicle, $h_l$ is the distance between the original center of the forklift and the roll center, $v_y$ and $v_x$ are the longitudinal velocity and the lateral velocity at the center of gravity of the vehicle, respectively, $\omega_r$ is the yaw rate of the forklift, $\beta$ is the sideslip angle of the forklift, $\psi$ is the body roll angle, $p$ is the roll velocity, $\dot{p}$ is the roll acceleration, $K_\psi$ is the total roll rate, $C_\psi$ is the roll damping, $I_x$ and $I_z$ are the moment of inertia around the $x$-axis and the $z$-axis, respectively, $I_{xz}$ is the product of $I_x$ and $I_z$, $k_1$, and $k_2$ are the front and the rear tire cornering stiffness values, respectively; $a$ and $b$ represent the distances from the center of gravity to the front and the rear axles, respectively. Additionally, $F_{yli}$ and $F_{yri}$, denote the left and right lateral tire forces, respectively, and $i$ represents the front and the rear. $\alpha$, which is directly proportional to $F$ ($F_{ylf}$, $F_{ylr}$, $F_{yrf}$, $F_{yrr}$, $F_{yri}$), denotes the sideslip angle of a tire.

As $\beta$ is always small enough, it can be approximated as

$$
\beta \approx \tan \beta = \frac{v_y}{v_x}
\tag{2}
$$

Based on this, sideslip angles of the front and rear tires can be approximated as

$$
\alpha_f = \beta + \frac{a}{v_x} \omega_r - \delta_f - R_f \psi
\tag{3}
$$

$$
\alpha_r = \beta - \frac{b}{v_x} \omega_r - R_r \psi
\tag{4}
$$

where $\delta_f$ denotes the front steering angle, and $R_f$ and $R_r$ are the roll steer coefficients of the front and rear wheels, respectively.

According to (1)–(4), the state space of a forklift can be derived as follows by considering $[\omega_r, \beta, \psi, p]$ as state vectors and $u(t) = \delta_f$ as the input vector [5]:

$$
\begin{cases}
\dot{x}(t) = Ax(t) + Bu(t) \\
y(t) = Cx(t)
\end{cases}
\tag{5}
$$

where

$$A = M_1^{-1}M_2 \quad B = M_1^{-1}M_3 \quad C = \begin{bmatrix} 1 & 0 & 0 & 0 \\ 0 & 1 & 0 & 0 \\ 0 & 0 & 1 & 0 \\ 0 & 0 & 0 & 1 \end{bmatrix}$$

$$M_1 = \begin{bmatrix} 0 & Mv_x & 0 & -M_sh_l \\ I_z & 0 & 0 & -I_{xz} \\ -I_{xz} & -M_sh_lv_x & 0 & I_x \\ 0 & 0 & 1 & 0 \end{bmatrix} \quad M_3 = \begin{bmatrix} k_1 \\ k_1a \\ 0 \\ 0 \end{bmatrix}$$

$$M_2 = \begin{bmatrix} \frac{-k_1a + k_2b}{v_x} - Mv_x & -k_1 - k_2 & (k_1R_f + k_2R_r) & 0 \\ -\frac{k_1a^2 + k_2b^2}{v_x} & -ak_1 + bk_2 & (k_1aR_f - k_2bR_r) & 0 \\ M_sh_lv_x & 0 & -(K_\psi - M_sgh_l) & -C_\psi \\ 0 & 0 & 0 & 1 \end{bmatrix}$$

## 2.2. Sensor Fault Model of a Forklift

Considering the external output disturbance and the cargo uncertainty, the equivalent sensor fault model of a forklift can be defined as (6) when sensor faults occur.

$$\begin{cases} \dot{x}(t) = Ax(t) + Bu(t) \\ y(t) = Cx(t) + Hd(t) + Ff \end{cases} \tag{6}$$

where $x(t)$ denotes the unmeasurable state vector, $u(t)$ denotes the measurable input vector, $y(t)$ denotes the measurable output vector, $d(t)$ denotes external output uncertainties and the nonlinear term, $f$ denotes the sensor fault, $H$ denotes the external interference matrix (provided), $F$ denotes the known distribution matrix of sensor faults; and $A$, $B$, and $C$ are constant matrices (also provided).

Assuming that multiple sensor faults occur, the model can be described as follows:

$$\begin{cases} \dot{x}(t) = Ax(t) + Bu(t) \\ y_1(t) = C_1x(t) + H_1d(t) + F_1f_1 \\ \quad \vdots \\ y_m(t) = C_mx(t) + H_md(t) + F_mf_m \end{cases} \tag{7}$$

where $m$ denotes the number of sensors that occur faults.

The sensor faults model (7) can be simplified as (8). Divide (7) into multiple single sensor fault models for sensor fault detection. When multiple sensor faults occur simultaneously, the corresponding part can be detected in detail to perform fault detection and reconstruction [20]:

$$\begin{cases} \dot{x}(t) = Ax(t) + Bu(t) \\ y_i(t) = C_ix(t) + H_id(t) + F_if_i \end{cases} \tag{8}$$

where $f_i$ denotes the $i^{th}$ sensor fault.

If state variable $z$ is defined as the first-order low-pass filtering output of $y(t)$ [21], the sensor fault can be transformed into an actuator fault:

$$\dot{z}_i = -A_{si}z_i + A_{si}y_i \tag{9}$$

where $A_{si}$ is the stable matrix of appropriate dimensions. Substituting (9) for (8), we can get

$$\begin{cases} \begin{bmatrix} \dot{x} \\ \dot{z}_i \end{bmatrix} = \begin{bmatrix} A & 0 \\ A_{si}C_i & -A_{si} \end{bmatrix} \begin{bmatrix} x \\ z_i \end{bmatrix} + \begin{bmatrix} B \\ A_{si}G_i \end{bmatrix} u + \begin{bmatrix} 0 \\ A_{si}H_i \end{bmatrix} \theta(t) + \begin{bmatrix} 0 \\ A_{si}F_i \end{bmatrix} f_i \\ z_i = \begin{bmatrix} 0 & I_1 \end{bmatrix} \begin{bmatrix} x \\ z_i \end{bmatrix} \end{cases} \tag{10}$$

A new state variable and the corresponding matrix can be expressed as follows:

$$\overline{x}_i = \begin{bmatrix} x & z_i \end{bmatrix}^T, \overline{y}_i = z_i, \overline{A}_i = \begin{bmatrix} A & 0 \\ A_{si}C_i & -A_{si} \end{bmatrix}, \overline{B}_i = \begin{bmatrix} B \\ A_{si}G_i \end{bmatrix}, \overline{C}_i = \begin{bmatrix} 0 & I \end{bmatrix}, \overline{H}_i = \begin{bmatrix} 0 \\ A_{si}H_i \end{bmatrix}, \overline{F}_i = \begin{bmatrix} 0 \\ A_{si}F_i \end{bmatrix}$$

Then, when the $i^{th}$ sensor fault occurs, the model can be expressed as follows:

$$\begin{cases} \dot{\overline{x}}_i(t) = \overline{A}_i\overline{x}_i(t) + \overline{B}_iu(t) + \overline{H}_id(t) + \overline{F}_if_i \\ \overline{y}_i(t) = \overline{C}_i\overline{x}_i(t) \end{cases} \tag{11}$$

Introduction of state variable z to Equation (11) converts the sensor fault into a pseudo-actuator fault. It forms an augmented state space system [2], and a single sensor fault model is constructed.

## 3. Design of the Adaptive Sliding Mode Observer

To design the corresponding ASMO, the following assumptions are proposed:

**Hypothesis 1.** *(A, C) is observable.*

**Hypothesis 2.** *Sensor faults are bounded matrices. It means there exists $\gamma_i > 0$, $\|f_i\| \leq \gamma_i$ and the value of $\gamma_i$ is unknown.*

**Hypothesis 3.** *The external disturbance is a bounded matrix. It means there exists $\varsigma > 0$, $\|\theta(t)\| \leq \gamma_i$.*

**Hypothesis 4.** *$(\overline{A}_i, \overline{C}_i)$ is observable, and there exists appropriate matrix $L_i, D_i, K_i$, positive definite matrix $P_i$ and $Q_i$, such that $(\overline{A}_i - L_i\overline{C}_i)$ remains stable, $P_i\overline{F}_i = \overline{C}_i^T D_i^T$, $P_i\overline{H}_i = \overline{C}_i^T K_i^T$ and $(\overline{A}_i - L_i\overline{C}_i)^T P_i + P_i(\overline{A}_i - L_i\overline{C}_i) = -Q_i$.*

**Hypothesis 5.** *$rank(\overline{C}_i\overline{F}_i) = rank(\overline{F}_i) = 1$. System (11) is a minimum phase system. It means the invariant zeros of the system are located on the left side of the complex plane. For all complex numbers satisfying $R(s) \geq 0$, (12) holds true.*

$$rank\begin{bmatrix} sI - \overline{A}_i & 0 & 0 \\ -A_{si}\overline{C}_i & sI + A_{si} & -A_{si} \\ 0 & I_1 & 0 \end{bmatrix} = rank\begin{bmatrix} sI & -\overline{A}_i \end{bmatrix} + 2 \tag{12}$$

Therefore, the invariant zeros of the system are the characteristic values of $\overline{A}_i$.

According to sensor fault model (11), the SMO can be described as [22]:

$$\begin{cases} \dot{\hat{\overline{x}}}_i(t) = \overline{A}_i\hat{\overline{x}}_i(t) + \overline{B}u(t) + L_i(\overline{y}_i - \hat{\overline{y}}_i) + \overline{F}_iv_i \\ \hat{\overline{y}}_i(t) = \overline{C}_i\hat{\overline{x}}_i(t) \end{cases} \tag{13}$$

where $\hat{\bar{x}}_i(t)$ is the state observation vector of $\bar{x}_i(t)$, $L_i$ is the gain matrix of the observer to be designed, $e_{yi} = \bar{y}_i - \hat{\bar{y}}_i$ is the output estimation error, $v_i$ is the discontinuous sliding mode control input vector which is used to cut off the effect of fault $f_i$, and it is expressed as follows:

$$
v_i = \begin{cases} \rho_i(t) \dfrac{D_i \bar{e}_{yi}}{\|D_i \bar{e}_{yi}\|} & \bar{e}_{yi} \neq 0 \\ 0 & \bar{e}_{yi} = 0 \end{cases}
\tag{14}
$$

where $\rho_i(t)$ denotes the adjustable gain parameter with the advantage of adaptive adjustment; it is designed using the corresponding adaptive algorithm without knowing the upper limit of the unknown sensor fault. The adaptive algorithm is expressed as follows:

$$
\frac{\mathrm{d}\rho_i(t)}{\mathrm{d}t} = \eta_i \|D_i \bar{e}_{yi}\| \mathrm{sgn}(\|D_i \bar{e}_{yi}\| - \lambda_i), i = 1, \cdots m
\tag{15}
$$

$$
\rho_i(0) > 0, \eta_i < 2, 0 < \lambda_i < 1, i = 1, \cdots m
\tag{16}
$$

According to (14), when $\mathrm{sgn}(\|D_i \bar{e}_{yi}\| - \lambda_i) = 1$, adjustable gain parameter $\rho_i(t)$ increases, and the increase rate is proportional to $\|D_i \bar{e}_{yi}\|$. Furthermore, when $\mathrm{sgn}(\|D_i \bar{e}_{yi}\| - \lambda_i) = -1$, $\rho_i(t)$ decreases. It means that once a sliding mode motion occurs, which means $\|D_i \bar{e}_{yi}\|$ is close to zero, $\|D_i \bar{e}_{yi}\| \leq \lambda_i$ when $\lambda_i$ takes a minor constant. Then, the adjustable gain parameter changes a little, which ensures that the sliding mode adjustable parameter is not too large, so that the chattering phenomenon caused by the excessive gain parameter can be weakened to some extent.

**Lemma 1.** *The adaptive adjustment law of the adjustable gain parameter in (14) should observe the following condition: the upper limit of $\rho_i(t)$ is $\rho_i^*$ and $\rho_i^* > \gamma_i$* [23].

The state estimation error is defined as follows:

$$
\bar{e}_i = \bar{x}_i - \hat{\bar{x}}_i
\tag{17}
$$

By subtracting Equation (12) from Equation (11) when the $i^{th}$ sensor fails, the deviation equation of the $i^{th}$ observer is:

$$
\dot{\bar{e}}_i = (\bar{A}_i - L_i \bar{C}_i) \bar{e}_i + \bar{H}_i d(t) + \bar{F}_i (f_i - v_i)
\tag{18}
$$

**Theorem 1.** *Based on the above definitions, when the $i^{th}$ sensor fails and the following conditions are met:*

*(1)* $\|\bar{e}_i\| > 2\|P_i\|\|\bar{H}_i\|\omega / \lambda_{\min}(Q_i)$

*(2) LMI:* $\begin{bmatrix} (\bar{A}_i - L_i \bar{C}_i)^T P_i + P_i (\bar{A}_i - L_i \bar{C}_i) & 0 \\ 0 & -P_i \end{bmatrix} \leq 0,$

*the dynamic systems of the state estimation error are asymptotically stable.*

**Proof.** The Lyapunov function is defined as follows:

$$
V(\bar{e}_i) = \bar{e}_i^T P \bar{e}_i + \frac{1}{2}(\rho_i - \rho_i^*)^2
\tag{19}
$$

and its derivative is calculated as:

$$
\begin{aligned}
\dot{V}(\bar{e}_i) &= \bar{e}_i^T\Big[\big(\overline{A}_i - L_i\overline{C}_i\big)^T P_i + P_i\big(\overline{A}_i - L_i\overline{C}_i\big)\Big]\bar{e}_i \\
&\quad +2\bar{e}_i^T P_i \overline{F}_i(f_i - v_i) + \dot{\rho}_i(\rho_i - \rho_i^*) + 2\bar{e}_i^T P_i \overline{H}_i d(t) \\
&= -\|\bar{e}_i\|\Big[\lambda_{\min}(Q_i)\|\bar{e}_i\| - 2\|P_i\|\|\overline{H}_i\|\omega\Big] \\
&\quad +2\bar{e}_i^T P_i \overline{F}_i(f_i - v_i) + \eta_i(\rho_i - \rho_i^*)\|D_i\bar{e}_{yi}\|\mathrm{sgn}(\|D_i\bar{e}_{yi}\| - \lambda_i) \\
&\leq -\|\bar{e}_i\|\Big[\lambda_{\min}(Q_i)\|\bar{e}_i\| - 2\|P_i\|\|\overline{H}_i\|\omega\Big] \\
&\quad +\big(2\gamma_i - 2\rho_i + \eta_i(\rho_i - \rho_i^*)\big)\|D_i\bar{e}_{yi}\| \\
&= -\|\bar{e}_i\|\Big[\lambda_{\min}(Q_i)\|\bar{e}_i\| - 2\|P_i\|\|\overline{H}_i\|\omega\Big] \\
&\quad +\big(2\gamma_i - 2\rho_i + 2\rho_i^* - 2\rho_i^* + \eta_i(\rho_i - \rho_i^*)\big)\|D_i\bar{e}_{yi}\| \\
&= -\|\bar{e}_i\|\Big[\lambda_{\min}(Q_i)\|\bar{e}_i\| - 2\|P_i\|\|\overline{H}_i\|\omega\Big] \\
&\quad +\Big[2(\gamma_i - \rho_i^*) - |\rho_i - \rho_i^*|(\eta_i - 2)\Big]\|D_i\bar{e}_{yi}\|
\end{aligned}
\tag{20}
$$

Combined with the condition in Theorem 1, $\dot{V}(\bar{e}_i) < 0$ and $\|\bar{e}_i\|$ converges in the following domain:

$$
U(\bar{e}_i) = \big\{\|\bar{e}_i\| \leq 2\|P_i\|\|\overline{H}_i\|\omega / \lambda_{\min}(Q_i)\big\}
\tag{21}
$$

## 4. Design of the Sensor Fault Reconstruction

The ASMO-based state estimation of the dynamic system is the purpose of reconstruction. In fact, the state variable will reach the sliding mode surface in limited time and stay there. There exists the following transformation matrix:

$$
T \in R^{n+1} = \begin{bmatrix} (\overline{C}_i)_{\perp}^T P_i \\ \overline{C}_i \end{bmatrix}
\tag{22}
$$

where $(\overline{C}_i)_{\perp} \in R^{(n+1)\times n}$ is a nonsingular matrix and the null space of $\overline{C}_i$. Therefore, $(\overline{C}_i)_{\perp}$ is the orthogonal complement space of $\overline{C}_i^T$. Then,

$$
T\bar{e}_i = \begin{pmatrix} e_{1i} \\ e_{yi} \end{pmatrix} = \widetilde{e}_i
\tag{23}
$$

The state estimation error under the new coordinates is

$$
\dot{\widetilde{e}}_i = T(\widetilde{A}_i)T^{-1}\widetilde{e}_i + T\overline{F}_i(f_{si} - v_i) + T\overline{H}_i\theta(t)
\tag{24}
$$

According to transformation matrix $T$ and $\overline{C}_i((\overline{C}_i)_{\perp}) = 0$, the following is true:

$$
T(\widetilde{A}_i)T = \begin{bmatrix} \widetilde{A}_{11i} & \widetilde{A}_{12i} \\ \widetilde{A}_{21i} & \widetilde{A}_{22i} \end{bmatrix}
\tag{25}
$$

$$
T\overline{F}_i = \begin{bmatrix} 0 \\ \overline{C}_i P_i^{-1}\overline{C}_i^T D_i^T \end{bmatrix}
\tag{26}
$$

$$
T\overline{F}_i v_i = \begin{bmatrix} 0 \\ v_{2i} \end{bmatrix}
\tag{27}
$$

$$
\begin{aligned}
v_{2i} &= \rho_i \overline{C}_i P_i^{-1}\overline{C}_i^T D_i^T \frac{D_i\bar{e}_{yi}}{\|D_i\bar{e}_{yi}\|} \\
&= \overline{C}_i \overline{F}_i v(t)
\end{aligned}
\tag{28}
$$

$$TH_i = \begin{bmatrix} 0 \\ \overline{C}_i P_i^{-1} \overline{C}_i^T K_i^T \end{bmatrix} \tag{29}$$

Under the new coordinates, (24) can be divided as follows:

$$\dot{\widetilde{e}}_{1i} = \widetilde{A}_{11i} e_{1i} + \widetilde{A}_{12i} e_{yi} \tag{30}$$

$$\begin{aligned} \dot{\widetilde{e}}_{yi} &= \widetilde{A}_{21i} e_{1i} + \widetilde{A}_{22i} e_{yi} + \overline{C}_i P_i^{-1} \overline{C}_i^T K_i^T d(t) \\ &+ \overline{C}_i P_i^{-1} \overline{C}_i^T D_i^T f_i(t) - \overline{C}_i P_i^{-1} \overline{C}_i^T D_i^T v_i(t) \end{aligned} \tag{31}$$

If the states of the error system arrive at the sliding mode surface in the finite time, which means $\dot{\widetilde{e}}_{yi} = e_{yi} = 0$, (30) and (31) can be transformed to the following:

$$\dot{\widetilde{e}}_{1i} = \widetilde{A}_{11i} e_{1i} \tag{32}$$

$$\begin{aligned} 0 &= \widetilde{A}_{21i} e_{1i} + \overline{C}_i P_i^{-1} \overline{C}_i^T D_i^T f_i(t) \\ &- \overline{C}_i P_i^{-1} \overline{C}_i^T D_i^T v_i(t) + \overline{C}_i P_i^{-1} \overline{C}_i^T K_i^T d(t) \end{aligned} \tag{33}$$

According to $P_i \overline{F}_i = \overline{C}_i^T D_i^T$, $e_{1i} \to 0$, we obtain the following:

$$\overline{C}_i \overline{F}_i f_i(t) \approx \overline{C}_i \overline{F}_i v_i(t) - \overline{C}_i \overline{H}_i d(t) \tag{34}$$

As $rank(\overline{C}_i \overline{F}_i) = 1$, and there exists the following pseudo inverse,

$$(\overline{C}_i \overline{F}_i)^+ = \left[ (\overline{C}_i \overline{F}_i)^T (\overline{C}_i \overline{F}_i) \right]^{-1} (\overline{C}_i \overline{F}_i)^T \tag{35}$$

$$W = -(\overline{C}_i \overline{F}_i)^+ \overline{C}_i \overline{H}_i \tag{36}$$

$f_{si}(t)$ can be reconstructed as

$$f_i(t) \to \left[ (\overline{C}_i \overline{F}_i)^+ (\overline{C}_i \overline{F}_i) \right] v_i(t) + W d(t) \tag{37}$$

In order to reduce chattering of the sliding mode, the sliding mode control input vector can be adjusted as follows:

$$v_i = \begin{cases} \rho_i(t) \dfrac{D_i \overline{e}_{yi}}{\|D_i \overline{e}_{yi}\| + \delta_i} & ; \quad if \quad \overline{e}_{yi} \neq 0 \\ 0 & ; \quad if \quad \overline{e}_{yi} = 0 \end{cases} \tag{38}$$

where $\delta_i$ is a small positive constant.

The fault value can be approximated as

$$f_i(t) \to \rho_i \left[ (\overline{C}_i \overline{F}_i)^+ (\overline{C}_i \overline{F}_i) \right] \dfrac{D_i \overline{e}_{yi}}{\|D_i \overline{e}_{yi}\| + \delta_i} - (\overline{C}_i \overline{F}_i)^+ \overline{C}_i \overline{H}_i d(t) \tag{39}$$

## 5. Improvement for Fault Reconstruction

To separate the sensor fault from the external output, (26) and (29) can be changed to (40) and (41) separately.

$$\widetilde{\overline{F}}_i = T \overline{F}_i = \begin{bmatrix} 0 \\ \overline{C}_i P_i^{-1} \overline{C}_i^T D_i^T \end{bmatrix} = \begin{bmatrix} \widetilde{\overline{F}}_{i1} \\ \widetilde{\overline{F}}_{i2} \end{bmatrix} \tag{40}$$

$$\widetilde{\overline{H}}_i = T\widetilde{H}_i = \left[ \begin{array}{c} 0 \\ \overline{C}_i P_i^{-1} \overline{C}_i^T K_i^T \end{array} \right] = \left[ \begin{array}{c} \widetilde{\overline{H}}_{i1} \\ \widetilde{\overline{H}}_{i2} \end{array} \right] \tag{41}$$

**Hypothesis 6.** *There exists a nonsingular matrix* $\mathbb{R}$, *which satisfies*

$$\mathbb{R}\left[ \begin{array}{cc} \widetilde{\overline{H}}_{i2} & \widetilde{\overline{F}}_{i2} \end{array} \right] = \left( \begin{array}{cc} \overline{E}_1 & \overline{E}_2 \\ 0 & \overline{E}_3 \end{array} \right) \tag{42}$$

where $\overline{E}_3$ is a nonsingular matrix.

**Lemma 2.** *Hypothesis (6) enables that the sensor fault reconstruction value can be separated from the external output, and makes accurate reconstruction of sensor faults possible. It is equivalent to the row operation which makes* $\left[ \begin{array}{cc} \widetilde{\overline{H}}_{i2} & \widetilde{\overline{F}}_{i2} \end{array} \right]$ *into the upper triangular matrix.*

According to (30) and (31), multiple (31) with nonsingular matrix $\mathbb{R}$, we obtain the following:

$$\begin{aligned} \mathbb{R}\dot{\tilde{e}}_{yi} = \ & \mathbb{R}\widetilde{A}_{21i} e_{1i} + \mathbb{R}\widetilde{A}_{22i} e_{yi} + \\ & \left( \begin{array}{cc} \overline{E}_1 & \overline{E}_2 \\ 0 & \overline{E}_3 \end{array} \right) \left( \begin{array}{c} d(t) \\ f_i(t) \end{array} \right) - \mathbb{R}\overline{C}_i P_i^{-1} \overline{C}_i^T D_i^T v_i(t) \end{aligned} \tag{43}$$

If the states arrive at the sliding mode surface in the finite time, which means $\dot{\tilde{e}}_{yi} = e_{yi} = 0$, (43) can be changed to

$$\mathbb{R}\overline{C}_i P_i^{-1} \overline{C}_i^T D_i^T v_i(t) \rightarrow \left( \begin{array}{cc} \overline{E}_1 & \overline{E}_2 \\ 0 & \overline{E}_3 \end{array} \right) \left( \begin{array}{c} d(t) \\ f_i(t) \end{array} \right) \tag{44}$$

Therefore, according to (44) and (38), robust estimation can be achieved, and the value of the sensor fault can be reconstructed as follows:

$$\hat{f}_i(t) \rightarrow \rho_i \overline{E}_3^{-1} \mathbb{R}_2 \overline{C}_i \overline{F}_i \frac{D_i \overline{e}_{yi}}{\|D_i \overline{e}_{yi}\| + \delta_i} \tag{45}$$

## 6. Experimental Verification

We used a TFC35 electric forklift manufactured by Hefei Banyitong Science and Technology Development Co., Ltd. (Hefei, China) as the study object. The wheel speed sensor, the steering wheel sensor, the yaw rate sensor, and the roll rate sensor were installed during the experiment. The maximum wheel speed was used as the estimation of the vehicle speed. The experiment was carried out in a warehouse with 20% indoor relative humidity. The experimental environment is shown in Figure 2, and the main parameters of the forklift are shown in Table 1. The moving speed is set to 15 km/h. The external disturbance is $H = \left[ \begin{array}{cccc} 0.1 & 0 & 0 & 0.04 \end{array} \right]^T$, the output disturbance is $\theta(t) = \sin(2t)$, and the distribution matrix of sensor faults is F = C. The parameters of the ASMO are set as $\rho_1(0) = \rho_2(0) = 10$, $\eta_1 = \eta_2 = -100$, $\lambda_1 = \lambda_2 = 0.5$.

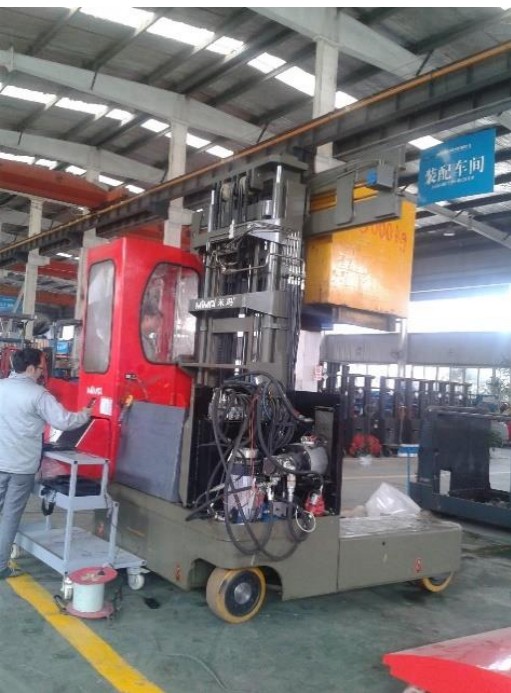

**Figure 2.** Experimental platform.

**Table 1.** Main parameters of the forklift.

| Symbol | Description | Values or Units |
|--------|-------------|-----------------|
| M | Mass of vehicle | 8720 kg |
| $M_s$ | Sprung mass | 7500 kg |
| $I_z$ | Yaw rate of inertia around the z-axis | 3650 kg·m$^2$ |
| $I_x$ | Yaw rate of inertia around the x-axis | 1800 kg·m$^2$ |
| $I_{xz}$ | Products of inertia around the x-axis and the z-axis | 500 kg·m$^2$ |
| $K_\psi$ | Total roll rate | 65,690 Nm/rad |
| $C_\psi$ | Roll damping | 2100 Nm/(rad/s) |
| $h_l$ | Distance from the center of gravity to the roll center | 0.3 m |
| a | Distance from the center of gravity to the rear axle | 1.2 m |
| b | Distance from the center of gravity to the front axle | 0.6 m |
| $k_1$ | Corning stiffness of the front axle | 85,000 N/rad |
| $k_2$ | Corning stiffness of the rear axle | 55,000 N/rad |
| $R_f$ | Roll-steering parameter of the front axle | 0.07 |
| $R_r$ | Roll-steering parameter of the rear axle | 0.05 |

Assume that forklift sensor faults occur in different feedback channels: yaw rate sensor faults and roll rate sensor faults. The expressions are as follows:

$$f_{s1} = \begin{cases} 3, & 12s \leq t < 17s \\ 0, & others \end{cases} \tag{46}$$

$$f_{s2} = \begin{cases} 0.2(t-10), & 10s \leq t < 13s \\ 0.6, & 13s \leq t < 15s \\ 0, & others \end{cases} \tag{47}$$

The double-lane-change (DLC) operating condition is often used to test lateral stability of vehicles. The experiment was conducted with a human operator, and the wheel angle sensor was installed to measure the steering wheel angle. The steering angle of the front wheel is shown in Figure 3.

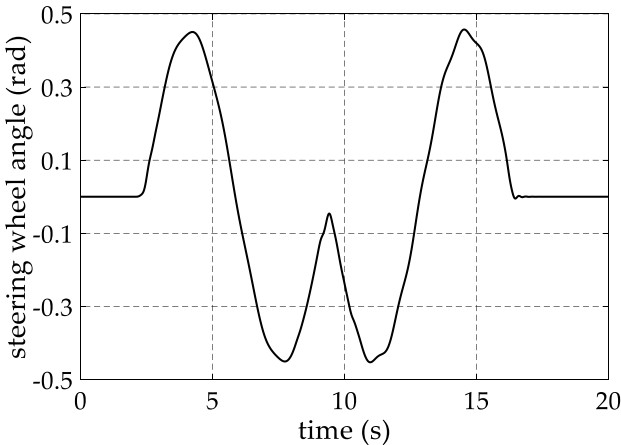

**Figure 3.** Steering wheel angle.

Not considering the output disturbance, the signals of the yaw rate sensor and the roll rate sensor without faults are shown in Figure 4.

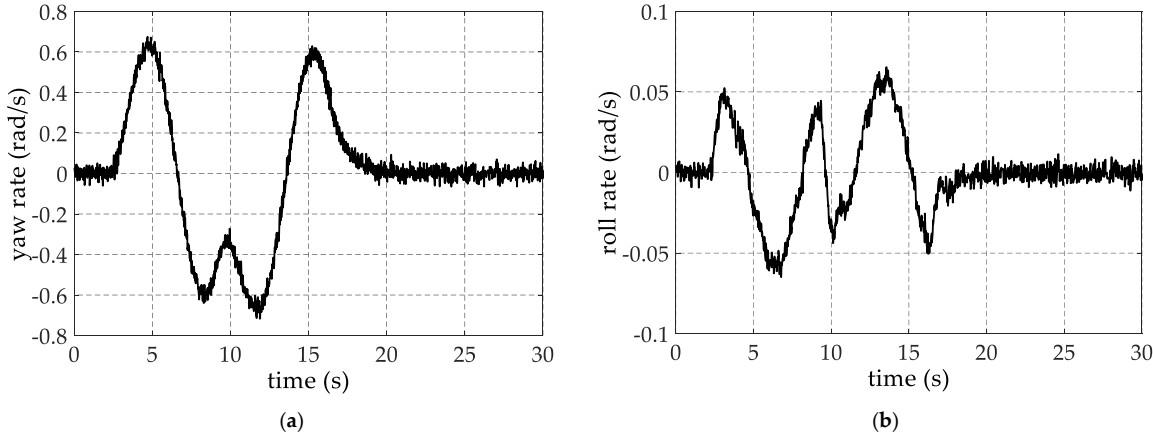

(**a**)                                                                 (**b**)

**Figure 4.** (**a**) The yaw rate sensor signal without a fault; (**b**) The roll rate sensor signal without a fault.

Not considering the output disturbance, Figure 5a shows the fault reconstruction value and the fault true value of the yaw rate sensor, and Figure 5b shows values of the roll rate sensor. The observer gives the fault reconstruction value of sensors during faults and the true value of the forklift when no faults take place. To compare the effectiveness of the ASMO, an SMO proposed in [23] is also shown in this manuscript. As the parameter of the SMO is determined by the upper bound of unknown faults, the error value of the SMO reconstruction value is larger than the ASMO. It means that without knowing the upper bound of faults, the parameter of the SMO cannot be determined precisely, and the result is also not as good as the ASMO. It is clear that an ASMO can detect faults of yaw rate and roll rate sensors without knowing the upper bound of faults.

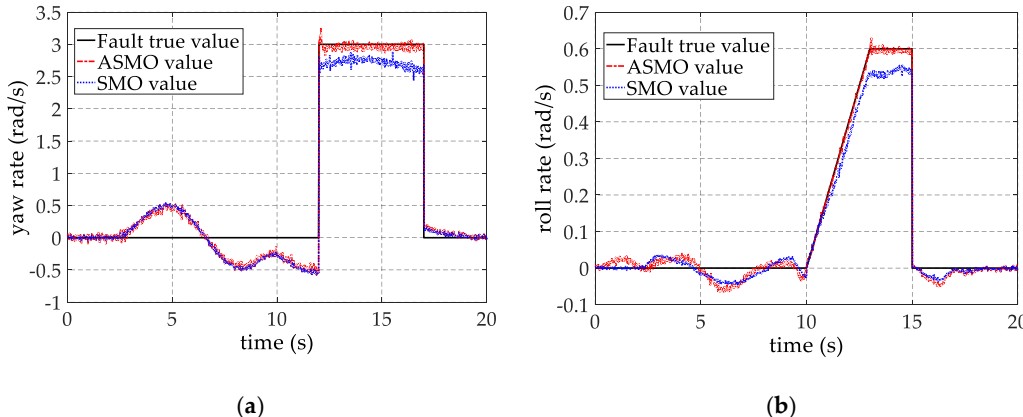

**Figure 5.** (**a**) Fault and its reconstruction value of the yaw rate sensor; (**b**) Fault and its reconstruction value of the roll rate sensor. ASMO: Adaptive sliding mode observer; SMO: sliding mode observer.

As shown in Figure 6, compared with no output disturbance, the signals of the yaw rate sensor and the roll rate sensor with output disturbance produce high fluctuation and cannot accurately reflect forklift conditions.

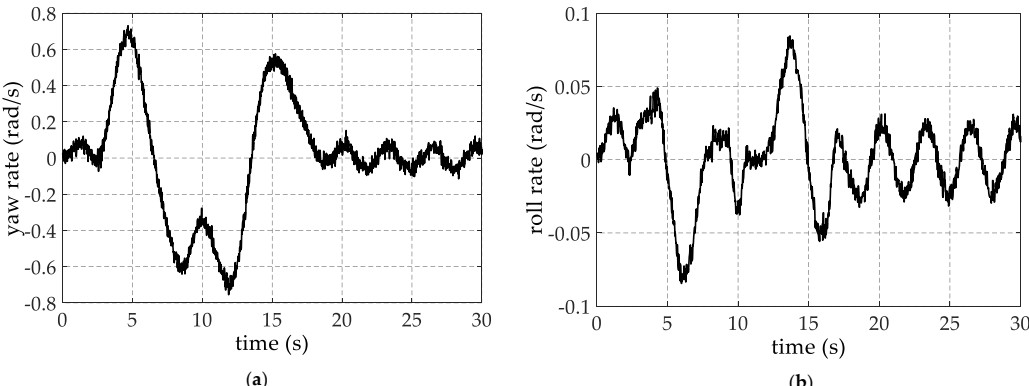

**Figure 6.** (**a**) The yaw rate sensor signal without a fault; (**b**) The roll rate sensor signal without a fault.

In Figure 7, compared with the SMO, the ASMO can reconstruct yaw rate sensor faults and roll rate sensor faults more effectively. On the other hand, the ASMO can also suppress external output disturbance.

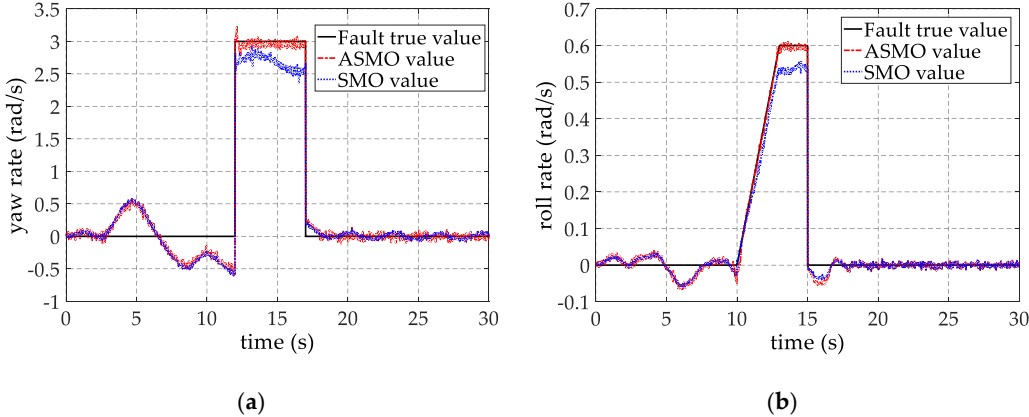

**Figure 7.** (**a**) Fault and its reconstruction value for the yaw rate sensor; (**b**) Fault and its reconstruction value for the roll rate sensor. ASMO: adaptive sliding mode observer; SMO: sliding mode observer.

As shown in Figure 8, based on the adaptive algorithm, adjustable gain parameters can change with the fault amplitude and the external output disturbance. When a fault occurs, values of the

adjustable parameters rise because of the increase in state estimation. And as $f_{s1}$ is higher than $f_{s1}$, $\rho_1$ is higher than $\rho_2$. When the fault is removed, the $\mathrm{sgn}(\|D_i \bar{e}_{yi}\| - \lambda_i) = -1$, $\rho_i(t)$ decreases correspondingly. It means that the ASMO can improve its fault-tolerant capability to deal with possible unknown faults.

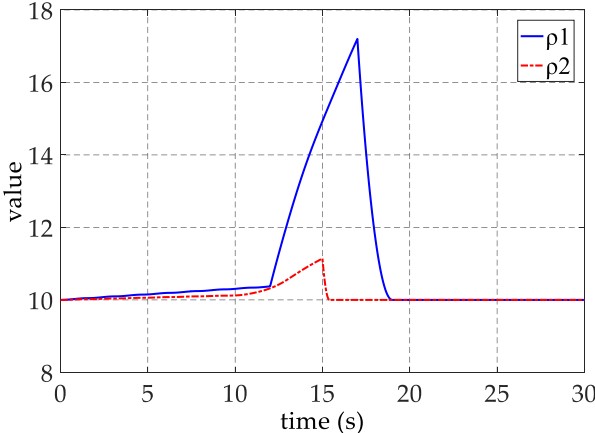

**Figure 8.** Adaptive update of the adjustable gain parameter.

## 7. Conclusions

To solve the problem of multi-sensor faults of forklifts, an ASMO-based sensor fault diagnosis method is proposed. First, for the 3-DOF model of electric forklifts, a linear state equation with output disturbance is constructed as its equivalent sensor fault model. By defining auxiliary state variables as output signal filters, sensor faults are transformed into actuator faults. Then, according to the fault model and the sliding mode control theory, an ASMO-based method of fault reconfiguration for multiple sensors is presented. Through the adaptive algorithm, multiple adjustable gain parameters of the ASMO can be adaptively adjusted with the change of fault amplitude and external disturbance without any prior assumptions. The comparison between the true value of a forklift fault and the reconstruction value shows that sensor faults can be reconstructed well. This verifies the effectiveness of the method.

Sensor fault reconstruction has become one of the most popular research topics in recent years. It provides the foundation and premise for the fault-tolerant control of electric vehicles. Based on the fault diagnosis, an FDI–FTC scheme was developed for an over-actuated electric vehicle with decentralized control in the presence of nonlinear disturbances to improve trajectory tracking of an autonomous vehicle in faulty situations [5]. Jia Qingxian et al. addressed the problem of integrated fault reconstruction and fault-tolerant control in linear systems subject to actuator faults via learning observers (LOs) [24]. A reconfigurable fault-tolerant controller was designed on the basis of the constructed LO to compensate for the influence of actuator faults by stabilizing the closed-loop system. All these papers provide the example of fault-tolerant control based on fault reconstruction for future studies.

**Author Contributions:** Z.Z. wrote the paper and helped in the experiment. B.X. conceived and designed the experiment, and helped in the writing. All authors have read and agreed to the published version of the manuscript.

**Funding:** This research was funded by the National Natural Science Foundation of China, grant number 51577046.

**Acknowledgments:** This work was supported by Hefei Banyitong Science and Technology Development Co., Ltd., and the authors are grateful to senior engineer Junliang Guo, Zijian Fang, Pengfei Li, Xingzhi Fang, and Weilin Lv for their cooperation and helpful suggestions. Furthermore, sincere thanks go to graduate students Junjie Huang, Chenggong Sun, Xu Zhang, and Zhizheng Jiang in the same study room.

**Conflicts of Interest:** The authors declare no conflict of interest.

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
