# Peer review of "Sensor Fault Reconstruction based on Adaptive Sliding Mode Observer for Forklift Fault-Tolerant Control System"

_applsci, doi:10.3390/app10041278_

Round 1

Reviewer 1 Report

There are some issues that should be addressed

It is similar to an existing previous research. It is recommended that the authors should describe the novelty and main constribution of the study based on the previous related studies. It would be better to describe the differences or add references if there is no strong novelty of the study.

- Equation (3)~(5) of below previous paper

Boulaabi, A. Sellami, and F. B. Hmida, “A Robust Sensor Fault Reconstruction Based on a New Sliding Mode Observer Design Method for Uncertain Delayed Systems: A Satellite System Application”, Circuits, System, and Signal ProcessingAI, Vol. 17, No. 3, pp. 88-97, 2015.

-Equation (8)~(9) of below previous paper

Rahme, and N. Meskin, “Adaptive Sliding Mode Observer for Sensor Fault Diagnosis of an Industrial Gas Turbine”, Control Engineering Practice, Vol. 38, No. 5, pp. 57-74, 2015.

In chapter 6, a fault was applied at a point where steering angle was close to zero. For reasonable evaluation, it’s better that fault application should be required at the time the state of folklift changes.

   3. In chapter 6, authors should give more explanations such the readers could understand the paper clearly. So, the authors should provide some comparison between the established method in this paper and other published approaches

   4. I wonder the reason that fault reconstruction value exists, even though fault true value is zero in figure of chapter 6

   5. Some further research topics could be discussed in the conclusion. Moreover, some relevant recent publications can be referred to give readers an up-to-date picture.

Reviewer 2 Report

The paper proposes a method of reconstruction of sensor faults based on second-order SMOs, applied to the Forklift. The observer is constructed from a 3-DOF Forklift model as well as a sensor model. The contribution of the paper lies in the fact that the characteristics of the observer and the particularity that the weight of the forklift cargo is changing at any time, based on the forklift sensor fault model with output disturbance, a SMO sensor fault reconstruction method with adaptive regulation law is proposed to solve the problem that some fault reconstruction methods need to know the upper bound value of the fault. 

The paper is well written and easy to understand, the models, although known in the literature, are well constructed and the equations are understandable. The experimental platform is described and the results show the effectiveness of the proposed approach.

However, the following points can be improved:

1) The introduction is well structured but can be enriched with other modeling and fault reconstruction techniques, such as for example the bond graph approach combined with a state model and an observer with unknown input for the FDI-FTC d an electric vehicle in Fault diagnosis and fault tolerent control of an electric Vehicle over-actuated", IEEE Transaction on Vehicular Technology; Vol. 62 (3), pp. 986 - 994, fev 2013. And hybrid approaches combining data-driven methods and SMO , for example Estimation of an incipient fault using an adaptive neurofuzzy sliding-mode observer. Energy and Buildings 77 (2014) 256–269.

2) What is the influence of the linearization of the model on the convergence of the observer.

3) The difference between an observer and an estimator is that the construction of an observer is associated with an analysis of stability or convergence of the observer. This part has not been sufficiently addressed in this work even so it is essential in the theoretical validation of an observer.

Round 2

Reviewer 1 Report

The authors revised the manuscript based on the reviewer reasonably. However, there are some issues that should be addressed for publication.

1. The authors described that the study is for forklift stability control system. However, it is not observed that the stability analysis of results for control system.

2. The experimental conditions should be described more clearly. Was the double lane change scenario conducted with human operator? or with controller? If the controller for DLC was used for test, the authors should explain about controller or human operator for clear understanding.

3. It is observed that the the magnitude of the injection term was not decreased after the fault was applied. It is recommended that the the authors should explain why the magnitude of the injection term is not decreased after the fault signal is removed.

4. The grid lines and titles(and y labels) should be applied to all figures for technical analysis and readability.

Round 3

Reviewer 1 Report

The authors revised the previous manuscript based on the comments.

- Throughout my reading, there are a number of typos. The authors are suggested to check for them carefully to improve the quality of the paper.
